# An Innovative Test for the Rapid Detection of Specific IgG Antibodies in Human Whole-Blood for the Diagnosis of *Opisthorchis viverrini* Infection

**DOI:** 10.3390/tropicalmed7100308

**Published:** 2022-10-17

**Authors:** Lakkhana Sadaow, Rutchanee Rodpai, Penchom Janwan, Patcharaporn Boonroumkaew, Oranuch Sanpool, Tongjit Thanchomnang, Hiroshi Yamasaki, Wannaporn Ittiprasert, Victoria H. Mann, Paul J. Brindley, Wanchai Maleewong, Pewpan M. Intapan

**Affiliations:** 1Department of Parasitology, Faculty of Medicine, Khon Kaen University, Khon Kaen 40002, Thailand; 2Mekong Health Science Research Institute, Khon Kaen University, Khon Kaen 40002, Thailand; 3Department of Medical Technology, School of Allied Health Sciences, Walailak University, Nakhon Si Thammarat 80161, Thailand; 4Faculty of Medicine, Mahasarakham University, Maha Sarakham 44000, Thailand; 5Department of Parasitology, National Institute of Infectious Diseases, Tokyo 162-8640, Japan; 6Department of Microbiology, Immunology and Tropical Medicine, Research Center for Neglected Diseases of Poverty, School of Medicine and Health Science, George Washington University, Washington, DC 20052, USA

**Keywords:** clonorchiasis, opisthorchiasis, serodiagnosis, IgG, point-of-care test, whole-blood sample, immunochromatographic test kit

## Abstract

Chronic human liver fluke infections caused by *Opisthorchis viverrini* and *Clonorchis sinensis* can last for decades and cause liver and biliary diseases, including life-threatening pathology prior to cholangiocarcinoma (CCA). CCA generally has a poor prognosis. Serological diagnosis can support parasitological examination in diagnosing disease and screening for the risk of CCA. Here, we present an improved and innovative lateral flow immunochromatographic test (ICT) kit that uses whole-blood samples (WBS) rather than serum to diagnose human opisthorchiasis, which also successfully diagnosed human clonorchiasis. This ICT includes a soluble worm extract of *O**. viverrini* adults and colloidal-gold-labeled conjugates of the IgG antibody to evaluate the diagnostic values with simulated WBS (*n =* 347). Simulated WBS were obtained by the spiking infection sera with red blood cells. The diagnostic sensitivity, specificity, positive and negative predictive values, and accuracy for detecting opisthorchiasis were 95.5%, 87.0%, 80.5%, 97.2%, and 90.1%, respectively. For clonorchiasis, these findings were 85.7%, 87.0%, 53.6%, 97.2%, and 86.8%, respectively. Combined for both diseases, they were 93.2%, 87.0%, 84.0%, 94.6%, and 89.6%, respectively. The ICT kit can possibly replace the ICT platforms for antibody detection in serum samples in field surveys in remote areas where sophisticated equipment is not available.

## 1. Introduction

The fish-borne liver flukes, *Clonorchis sinensis*, *Opisthorchis viverrini*, and *Opisthorchis felineus* are biological carcinogens [1]. Human infections with these flatworms have an associated morbidity and the risk of cholangiocarcinoma (CCA) in East and Southeast Asia [2,3,4,5]. Detection of the parasite egg in feces, bile or duodenal contents represents the gold standard for the diagnosis of human infection with these helminths. However, this approach is laborious and requires a proficient skilled technician and microscopist [2]. Serologic and molecular tools have been developed to support diagnosis and screening [6,7] of opisthorchiasis [8,9,10,11,12,13] and clonorchiasis [14,15,16,17,18,19]. Kits employing lateral flow immunochromatographic tests (ICT kits) have been developed for point-of-care (POC) diagnosis for opisthorchiasis and clonorchiasis [20,21]. An ICT for rapid detection of the anti *O**. viverrini* IgG antibody in human sera [20] showed promise as a potential biomarker predictive for CCA in regions endemic for opisthorchiasis [22]. Nonetheless, the aforementioned tools were optimized for serum and do not use whole-blood samples (WBS). Here, we improved the ICT kit of Phupiewkham et al. [21] for detection of the specific IgG antibody in simulated WBS using a soluble worm extract of *O**. viverrini* adults as the antigen. We termed this innovation the opisthorchiasis blood immunochromatographic test, OvB-test kit. The performance of the OvB-test kit was evaluated using simulated and anticoagulated/treated whole blood. The OvB-test kit reliably diagnosed human infections with *O**. viverrini* and also successfully diagnosed human clonorchiasis. This kit can support the clinical diagnosis of liver fluke infections in remote and field settings without the requirement for sophisticated laboratories or equipment.

## 2. Materials and Methods

### 2.1. Parasite and Antigen Preparations

*Opisthorchis viverrini* metacercariae were collected from naturally infected cyprinid fish and experimentally infected hamsters. Adult *O**. viverrini* worms were collected from the livers and bile ducts of hamsters at three months after infection. A soluble worm extract was prepared from adult-stage *O**. viverrini* worms as described by Intapan and Maleewong [23]. Briefly, the worms were homogenized with a tissue grinder in a small volume of 0.1 M phosphate-buffered saline (PBS), pH 7.4, containing proteinase inhibitors (cOmplete^TM^ ULTRA Tablets, Mini EASYpack Protease Inhibitor Cocktail Tablet, Roche, Basel, Switzerland). The lysate was subjected to sonication in an ultrasonic disintegrator (SONICS vibra cell VCX 500 Ultrasonic Liquid Processor, Sonics & Materials, Inc., Newtown, CT, USA), after which the suspension was clarified by centrifugation at 10,000× *g* for 30 min at 4 °C. The protein concentration of the antigen was determined according to the method proposed by Bradford [24] and aliquots of this antigen preparation were stored at −80 °C.

### 2.2. Preparation of Simulated WBS

The simulated WBS preparations were as described previously [25]. Briefly, red blood cells (RBCs) from leftover whole-blood samples (group O type), which were preserved in citrate phosphate dextrose adenine-1 anticoagulant, from healthy donors in the Central Blood Bank, Faculty of Medicine, Khon Kaen University, were used. These RBCs (0.5 mL) were pelleted by centrifugation at 13,200× *g* for 10 min at 4 °C; the packed RBCs were washed three times with PBS, pH 7.4, by centrifugation at 13,200× *g* for 10 min at 4 °C. Thereafter, the RBCs were re-suspended in PBS, pH 7.4, and the suspension was dispensed into 10 µL aliquots, which were then recentrifuged at 13,200× *g* for 10 min at 4 °C. The supernatant was discarded, and the packed RBCs (approximate 3.5 µL) were used to prepare the simulated WBS. Before performing the experiment, human serum (above) was spiked with the packed RBCs, 6.5 µL, to restore the simulated WBS to the equivalent normal range of human blood components and stored at 4 °C until used.

### 2.3. Clinical Samples

Serum samples were obtained from the frozen bank of the Department of Parasitology, Faculty of Medicine, Khon Kaen University. Clonorchiasis sera were provided by the Department of Parasitology, National Institute of Infectious Diseases, Tokyo, Japan. Samples of the sera were divided into four groups; (a) the control group (*n* = 40) from healthy volunteers from northeastern Thailand screened to be free of any parasite infections at that time of blood donation; (b) the proven opisthorchiasis group (*n* = 112) from persons infected with *O**. viverrini* in endemic areas in the northeastern provinces of Thailand; (c) the clonorchiasis group (*n* = 35) from persons infected with *C**. sinensis* in the Guangxi Zhuang Autonomous Region, China, an endemic locality [26] and (d) an “other” group (*n* = 160) from cases infected with other parasites. Stool samples from healthy volunteers, opisthorchiasis, paragonimiasis, taeniasis saginata, ascariasis, hookworm infections, trichuriasis, capillariasis philippinensis, strongyloidiasis, giardiasis, blastocystosis, and lecithodendriid fluke infections were confirmed the presence or absence of the parasites by using modified formalin ethyl acetate concentration method [27]. The modified Kato- Katz technique [28] and ELISA [26] were used to confirm clonorchiasis. Fascioliasis was confirmed by the recovery of worms and by serology [29], cysticercosis was confirmed by serology and computed tomography [30], gnathostomiasis by serological methods [31], angiostrongyliasis by serological methods and clinical presentation [32], trichinellosis by detection of intramuscular larvae and serological methods [33], and sparganosis by the detection of the plerocercoids (Table 1). Ethylenediamine tetraacetic acid (EDTA) anti-coagulated WBS from confirmed opisthorchiasis (*n* = 10) and non-infected individuals (*n* = 10), as well as the incidentally found parasitosis (strongyloidiasis (*n* = 10) and other parasitosis (*n* = 10) from field collections in Khon Kaen Province, northeastern Thailand), were also used (Appendix A).

### 2.4. Preparation of the Immunochromatographic Test

The components of the ICT (Figure 1A) were laminated in five layers: (1) modified backing card (paper lower cassette), (2) nitrocellulose membrane (Sartorius Stedim Biotech SA, Göttingen, Germany) of the antigen line (T-line) and anti-mouse IgG antibody (Lampire Biological Laboratories, Pipersville, PA, USA) line (C-line), (3) conjugated pad of antibody-labeled gold nanoparticle, (4) sample pad (Millipore, Massachusetts, MA, USA) and (5) absorbent pad (Whatman Schleicher & Schuell, Dassel, Germany). The laminate was cut into strips of 0.5 cm length using a guillotine (BioDot, Irvine, CA, USA). Finally, the test strip laminate was inserted into a plastic cassette cartridge, and the cassette was closed with a cover (Adtec Inc., Oita, Japan) (Figure 1). The nitrocellulose membrane was coated with the *O**. viverrini* antigen at the T-line and the anti-mouse IgG antibody at the C-line, using the XYZ3210 Dispense Platform (BioDot), at a flow rate of 0.1 µL/mm. The monoclonal anti-human IgG antibody-labeled gold nanoparticles (Kestrel BioSciences Co., Pathumthani, Thailand) were sprayed onto a glass microfiber filter (Whatman Schleicher and Schuell) at a flow rate of 1 µL/mm to produce the conjugated pad.

### 2.5. Immunochromatographic Testing Method

The paper cassettes were optimized for antibody detection in the simulative WBS. Briefly, simulative WBS were diluted in 1:10 (0.5 µL of the blood sample plus 4.5 µL of running buffer (0.5% casein in 25 mM Tris-HCl, pH 8.0)) and were applied onto a sample well (S). Next, 60 µL of the running buffer was applied onto the buffer well. The kit results were read at 15 min with the naked eye and judged as positive or negative by reference to a color card, with the cutoff for positive defined as ≥ 0.5 (Figure 1B). The appearance of the red bands at the T-line and the C-line were judged as positive, whereas only the C-line appeared in negative cases. In the absence of the red band at the C-line, the test was determined to be invalid. For samples from the field-based studies, 0.5 µL of the EDTA anticoagulated WBS was used instead of the simulative blood samples.

In addition, the OvB-test kit was found to remain active and stable for 12 months at 25 °C or 4 °C under dry and dark conditions.

### 2.6. Data Analysis

The diagnostic values of the OvB-test kit were evaluated based on the specificity, sensitivity, positive predictive value (PPV), negative predictive value (NPV), and accuracy using the parameters: true negative (TN), false positive (FP), true positive (TP), and false negative (FN). All datasets were calculated according to the following formulas: specificity (%) = 100 × [(TN)/(TN + FP)], sensitivity (%) = 100 × [(TP/(TP + FN)], PPV (%) = 100 × [TP/(TP + FP)], NPV (%) = 100 × [TN/(TN + FN)], and accuracy (%) = 100 × [TN+ TP/total]. TN was the number of control samples (other parasitosis and healthy controls) that were negative by the assay. TP was the number of proven opisthorchiasis, clonorchiasis, or both samples that were positive by the assay. FP was the number of control samples that were positive by the assay, and FN was the number of proven opisthorchiasis, clonorchiasis or both samples that were negative by the assay. Stata version 13.1 was used to analyze the positive and negative likelihood ratios: the former was calculated as sensitivity/(100−specificity) and the latter was calculated as (100−sensitivity)/specificity. The analyses reported here were performed using the criteria of the STARD 2015 list for reporting diagnostic accuracy studies [34].

## 3. Results

The kit was improved to detect the specific IgG antibody using simulative WBS for diagnosis opisthorchiasis and possible clonorchiasis (Table 1). None of the 40 healthy humans control showed a positive result. In total, 107 of 112 opisthorchiasis cases and 30 of 35 clonorchiasis cases were positive, respectively. Some cross-reaction with other parasitic diseases was seen; there were 26 out of 160 cases: fascioliasis (*n* = 7), paragonimiasis (*n* = 6), lecithodendriid fluke infections (*n* = 4), cysticercosis (*n* = 1), sparganosis (*n* = 1), angiostrongyliasis (*n* = 1), hookworm (*n* = 1), trichinosis (*n* = 3), blastocystosis (*n* = 1) and giardiasis (*n* = 1) (Table 1, Figure 2). For the field studies, there were 10 EDTA-anticoagulated WBS from known opisthorchiasis cases, and all were positive in the kit, whereas the 10 WBS from healthy volunteers were all negative. Two strongyloidiasis cases were positive, as was one mixed case with blastocystosis and lecithodendriid fluke infection, although the intensities were only weakly or borderline positive (Appendix A).

The diagnostic values and the intensities of the positive bands of the kit were evaluated for opisthorchiasis, clonorchiasis, and the combined cases (Table 2, Figure 2). For opisthorchiasis, the sensitivity, specificity, and positive and negative predictive values were 95.5% (95% CI: 89.9–98.5%), 87.0% (95% CI: 81.5–91.3%), 80.5% (95% CI: 72.7–86.8%), and 97.2% (95% CI: 93.6–99.1%), respectively. For clonorchiasis, the sensitivity, specificity, and positive and negative predictive values were 85.7% (95% CI: 69.7–95.2%), 87.0% (95% CI: 81.5–91.3%), 53.6% (95% CI: 39.7–67.0%), and 97.2% (95% CI: 93.6–99.1), respectively. When opisthorchiasis and clonorchiasis were combined for evaluation as a disease entity, the sensitivity, specificity, and positive and negative predictive values were 93.2% (95% CI: 87.8–96.7%), 87.0% (95% CI: 81.5–91.3%), 84.0% (95% CI: 77.5–89.3%), and 94.6% (95% CI: 90.2–97.4%), respectively.

## 4. Discussion

Chromatographical lateral flow approaches are currently used as POC testing tools and have been found to be advantageous in a broad range of applications including diagnosis of infectious diseases [35,36]. These applications include the serodiagnosis of human opisthorchiasis and/or clonorchiasis and have shown potential utility as a biomarker to predict the risk of CCA in regions endemic for opisthorchiasis [20,21,22]. However, these tools have been developed for serum and do not utilize WBS [20,21,22]. In this study, the OvB-test kit was improved for successful detection of the specific IgG antibody in simulated WBS from human infection with *O**. viverrini* and also successfully diagnosed human clonorchiasis. This outcome was unsurprising given the close phylogenetic relationship between these fish-borne trematodes, *O**. viverrini* and *C**. sinensis*, and the epitopes shared by antigens of both flukes [14].

The diagnostic values of the OvB-test kit were compared for opisthorchiasis, clonorchiasis, and their combined cases (Table 2). The sensitivity, specificity, and positive and negative predictive values of the OvB-test kit were close together as with the findings of our earlier reports on the other ICT kits (OvES kit; [20]) and (OvSO-IgG kit; [21]). Both OvES and OvSO-IgG kits performed well in diagnosing opisthorchiasis, clonorchiasis, and their combined cases [20,21]. The variation in the diagnostic values of the present OvB-test kit and the two previous reports was possibly the result of the different conditions applied during the ICT kit optimization and the development of the evaluation between the IgG antibody detection in whole blood and sera.

The EDTA-anticoagulated WBS from opisthorchiasis patients from the field samples were all positive according to the ICT kit, while the 10 WBS from healthy volunteers were all negative (Appendix A). Therefore, our OvB-test kit can be used for serodiagnosis of opisthorchiasis and possible clonorchiasis at the bedside and in the field.

Our study had some limitations. First, the OvB-test kit showed cross-reactions with several other parasitic infections such as fascioliasis (7/10), paragonimiasis (6/10), cysticercosis (1/10), sparganosis (1/10), angiostrongyliasis (1/10), infections with lecithodendriid flukes (4/10) and hookworms (1/10), trichinellosis (3/10), giardiasis (1/10), and blastocystosis (1/10). This result may have occurred because the *O**. viverrini* soluble worm extract antigen included > 30 polypeptides [8] and consequently may share epitopes with other parasites. Moreover, subclinical infection with liver flukes may have associated with these findings because the human sera for evaluating the control specificity were mostly from people from northeast Thailand, a region endemic for opisthorchiasis [2]. Nevertheless, these limitations should not obstruct the application of this POC testing tool given the clinically different settings of opisthorchiasis and clonorchiasis [37,38,39,40,41,42,43,44]. Attention should be paid, however, to the possibility of *Fasciola* infection in the regions where fascioliasis co-exists with opisthorchiasis and clonorchiasis and to comprehensively diagnose the disease including other examination data, such as clinical signs, stool examination, radio-imaging, etc. [45]. Second, the IgG antibody detection by the ICT kit cannot be used for opisthorchiasis diagnosis of all fluke infection periods, as the antibody responses to *O**. viverrini* antigens are detected as early as 2 weeks after infection [46,47]. The rank of fluke-specific IgG was elevated in chronic opisthorchiasis [48]. A drawback of antibody-based detection methods is the inability to differentiate past and present infections with *O**. viverrini*, due to the long half-life of the antibody response to *O**. viverrini*, which can persist in the infected and non-infected hosts for years after the cure [49,50]. This evidence affects the resulting window period of the test in that our ICT test cannot be used to diagnose acute infection. Third, clinicians working in this endemic area should be reminded that there were differences in the reactions of detection, as the WBS from the field collections gave a band level between 0.5 and 2 (Appendix A), and the band level was between 0.5 and 5 in the simulated WBS reported in Table 1. These results may be due to sample collections from different periods of infections or the intensity of the helminthic infections in the populations studied.

## 5. Conclusions

To the best of our knowledge, no POC serodiagnostic kit for the detection of the anti-*Opisthorchis* antibody using WBS has been previously available. The present kit is the first diagnostic tool able to detect the specific IgG antibody in simulative WBS for serodiagnosis of human opisthorchiasis, possible clonorchiasis, and both diseases in human blood. This kit showed high sensitivity and specificity in the simulated WBS obtained by spiking patient serum with red blood cells. In addition, the EDTA-anticoagulated WBS was applied. A further advantage of this POC kit is that it can utilize of samples of fingerstick blood, thereby removing the need to draw venous blood and to separate the serum. Thus, the kit can replace the ICT platforms for antibody detection in sera. The kit is suitable to not only support clinical diagnosis at the bedside, but also for field-based surveys to predict the risk of CCA in endemic and remote regions, which may lack sophisticated instrumentation. This study indicates possible prognostic and diagnostic biomarkers for screening for the risk of CCA in regions endemic for liver fluke infection, and in disease prevention. However, this was an in vitro study with simulated WBS in a laboratory setting, more evaluation should be conducted in field studies.

## Figures and Tables

**Figure 1 tropicalmed-07-00308-f001:**
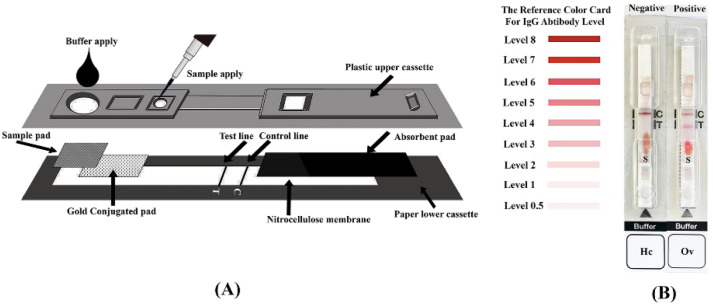
(**A**) Schematic illustration of the OvB-test kit for the blood sample, and (**B**) the positive and negative results are shown with a reference board. A red band appears at the C line in the negative case (Hc) and two bands appear at the C and T lines in the opisthorchiasis (Ov). The reference color card positive for the IgG antibody level is also shown (cutoff 0.5 or more).

**Figure 2 tropicalmed-07-00308-f002:**
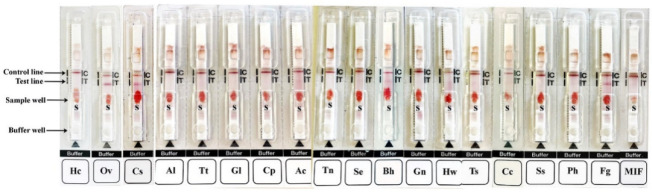
Representative results of the OvB- test kit using stimulated WBS from the healthy control (Hc), opisthorchiasis (Ov), clonorchiasis (Cs), and other parasitic diseases, including: (Al) ascariasis; (Tt) trichuriasis; (Gl) giardiasis; (Cp) capillariasis philippinensis; (Ac) angiostrongyliasis; (Tn) taeniasis saginata; (Se) sparganosis; (Bh) blastocystosis; (Gn) gnathostomiasis; (Hw) hookworm infections; (Ts) trichinosis; (Cc) cysticercosis; (Ss) strongyloidiasis; (Ph) paragonimiasis; (Fg) fascioliasis; and (MIF) minute intestinal fluke infections (lecithodendriid flukes). The intensity of the test line was visually estimated, according to the reference card. (S) indicates sample well.

**Table 1 tropicalmed-07-00308-t001:** Type of simulated whole-blood sample reactivities examined by the kit for the rapid diagnosis of human opisthorchiasis, clonorchiasis, and both diseases in human blood samples.

Type of Samples	Number of Positive/Total Samples Evaluated (Intensity Level of Bands ^a^)
Healthy volunteers	0/40
Opisthorchiasis	107/112 (0.5–5)
Lecithodendriid fluke infections	4/10 (1)
Clonorchiasis	30/35 (0.5–4)
Fascioliasis	7/10 (0.5–3)
Paragonimiasis	6/10 (0.5–2)
Cysticercosis	1/10 (0.5)
Taeniasis saginata	0/10
Sparganosis	1/10 (0.5)
Ascariasis	0/10
Angiostrongyliasis	1/10 (1)
Hookworm infections	1/10 (0.5)
Trichinellosis	3/10 (0.5–1)
Trichuriasis	0/10
Capillariasis philippinensis	0/10
Gnathostomiasis	0/10
Strongyloidiasis	0/10
Blastocystosis	1/10 (2)
Giardiasis	1/10 (2)
Total number of evaluated samples	347

^a^ The intensity level of band at the test-line was evaluated according to the reference card.

**Table 2 tropicalmed-07-00308-t002:** Statistical analysis of the diagnostic values of the kit.

Data	Opisthorchiasis (Ov) ^a^	Clonorchiasis (Cs) ^b^	Ov and Cs ^c^
Sensitivity (%)	95.5 (89.9–98.5)	85.7 (69.7–95.2)	93.2 (87.8–96.7)
Specificity (%)	87.0 (81.5–91.3)	87.0 (81.5–91.3)	87.0 (81.5–91.3)
Positive predictive value (%)	80.5 (72.7–86.8)	53.6 (39.7–67.0)	84.0 (77.5–89.3)
Negative predictive value (%)	97.2 (93.6–99.1)	97.2 (93.6–99.1)	94.6 (90.2–97.4)
Positive likelihood ratio	7.35 (5.12– 10.5)	6.59 (4.49–9.67)	7.17 (5.0–10.3)
Negative likelihood ratio	0.05 (0.02–0.12)	0.16 (0.07–0.37)	0.10 (0.04–0.14)
Accuracy (%)	90.1 (86.2–93.2)	86.8 (81.8–90.9)	89.6 (85.9–92.6)

^a^ Individual opisthorchiasis; ^b^ individual clonorchiasis; ^c^ a combination of opisthorchiasis and clonorchiasis; numbers in parentheses indicate a 95% confidence interval.

## Data Availability

Not applicable.

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
