# Peer review of "An Innovative Test for the Rapid Detection of Specific IgG Antibodies in Human Whole-Blood for the Diagnosis of Opisthorchis viverrini Infection"

_tropicalmed, 2022, doi:10.3390/tropicalmed7100308_

Round 1

Reviewer 1 Report

Comments

The manuscript is concise and written effectively. The materials and methods section is quite beneficial. However, I have issues with this work's central argument.

1. IgG and the stage of infection

IgG has been utilized as a biomarker for CCA because it may indicate the Ov antigen in the body, which is responsible for cacogenesis. Therefore, while discussing the infection stage, which in this case is Ov, the author should refer to the identification of acute, chronic, subsiding, or immunological phase based on IgG.

Will the IgG test kit detect the actual presentation of the worms in the human body, or will it represent antigen presentation regardless of whether or not the Ov is present?

IgG is the immunological marker used to measure CCA risk. I understand that the author attempted to mimic a scenario in which each specimen's infection status is confirmed. Therefore, the validity and performance of the kit may be evaluated. However, the author should be aware that this is not typical of an infection in which all infected individuals are at the same stage.

The experiment is engaging and instructive, but the author should be concerned and provide additional clarification, as I indicated for this issue.

I propose that referring to the kit as a tool for detecting Ov infection may be too optimistic for the reasons mentioned above.

2. Test kit for diagnostic testing

Did the author intend to use this kit as a screening or diagnostic test?

This should be addressed in the manuscript. I assumed that the test scenario was designed for an Ov endemic environment, that the serum was collected from an Ov and Cs endemic region, and that simulated blood was enriched with lab-grown Ov antigens. With such a high infection density, the test could be a good candidate for a diagnostic test, not to mention its practicability and feasibility, which made it highly convenient.

In many regions of Thailand, the risk of infection and eating undercooked fish remains high. Despite this, the intensity of the infection is relatively low, which could result in a somewhat low amount of Ov antigen. Therefore, using this kit as a screening test may be problematic.

I encourage the author to elaborate on this subject in the article.

3. Predictive value

Positive and negative predictive value indicators are presented in this work.

The positive predictive value (PPV) is determined similarly to the risk ratio. Therefore, PPV will be associated with infection prevalence. As if the test were extensively utilized, PPV is considered the determining criterion for assessing cost-effectiveness by public health professionals.

However, this infection was simulated; consequently, we could not determine the true prevalence of Ov and Cs infection. As such, the PPV may not reflect the actual value. In contrast to sensitivity and specificity, they would be technically equivalent at all levels of disease prevalence.

Again, this is not wrong; the author should address this issue as it is still the outcome of an in-vitro study.

4. Page 5

The explanations of sens/spec and PPV/NPV are redundant. The author could relocate the explanation to the appendix to shorten the document's main body.

Author Response

Point by point response to Reviewer 1 

Comments

The manuscript is concise and written effectively. The materials and methods section is quite beneficial. However, I have issues with this work's central argument.

1. IgG and the stage of infection

IgG has been utilized as a biomarker for CCA because it may indicate the Ov antigen in the body, which is responsible for cacogenesis. Therefore, while discussing the infection stage, which in this case is Ov, the author should refer to the identification of acute, chronic, subsiding, or immunological phase based on IgG.

Reply: Thank you for pointing these out; we have revised the discussion section, last paragraph. We added the sentence “Second, the IgG antibody detection by the ICT kit cannot be used for opisthorchiasis diagnosis of all fluke infection periods, as the antibody responses to O. viverrini antigens are detected as early as 2 weeks after infection [46-47]. The rank of fluke-specific IgG was elevated in chronic opisthorchiasis [48]. A drawback of antibody-based detection methods is the inability to differentiate past and present infections with O. viverrini, due to the long half-life of the antibody response to O. viverrini, which can persist in the infected and non-infected hosts for years after the cure [49-50]. This evidence affects the resulting window period of the test in that our ICT test cannot be used to diagnose acute infection ”.

  1.  Sirisinha, S.; Tuti, S.; Vichasri, S.; Tawatsin, A. Humoral immune responses in hamsters infected with Opisthorchis viverrini. Southeast Asian J. Trop. Med. Publ. Health 1983, 14, 243–251.
  2.  Sripa, B.; Kaewkes, S. Relationship between parasite-specific antibody responses and intensity of Opisthorchis viverrini infection in hamsters. Parasite Immunol. 2000, 22, 139–145. doi: 10.1046/j.1365-3024.2000.00286.x.
  3. Pinlaor, P.; Pongsamart, P.; Hongsrichan, N.; Sangka, A.; Srilunchang, T.; Mairiang, E.; Sithithaworn, P.; Pinlaor, S. Specific serum IgG, but not IgA, antibody against purified Opisthorchis viverrini antigen associated with hepatobiliary disease and cholangiocarcinoma. Parasitol Int. 2012, 61, 212-216. doi: 10.1016/j.parint.2011.06.014
  4.  Thammapalerd, N.; Tharavanij, S.; Nacapunchai, D.; Bunnag, D.; Radomyos, P.; Prasertsiriroj, V. Detection of antibodies against Opisthorchis viverrini in patients before and after treatment with praziquantel. Southeast Asian J. Trop. Med. Publ. Health 1988, 19, 101–108.
  5. Ruangsittichai, J.; Viyanant, V.; Vichasri-Grams, S.; Sobhon, P.; Tesana, S.; Upatham, E.S.; Hofmann, A.; Korge, G.; Grams, R. Opisthorchis viverrini: identification of a glycine-tyrosine rich eggshell protein and its potential as a diagnostic tool for human opisthorchiasis. Int. J. Parasitol. 2006, 36, 1329–1339. doi: 10.1016/j.ijpara.2006.06.012.

Please see the revised manuscript (discussion, last paragraph and ref no. 46 to 50).

Will the IgG test kit detect the actual presentation of the worms in the human body, or will it represent antigen presentation regardless of whether or not the Ov is present?

Reply: Our IgG test kit can detect IgG antibody against Opisthorchis antigen, but a drawback of antibody-based detection methods is the inability to differentiate past and present infections with O. viverrini, due to the long half-life of the antibody response to O. viverrini, which can persist in the infected and non-infected hosts for years after cure (Thammapalerd et al., 1988; Ruangsittichai et al., 2006; ref. no. 49-50). Nevertheless, cases that were positive for anti O. viverrini IgG antibody in human sera as shows promise as the potential biomarker predictive for CCA in regions endemic for opisthorchiasis (Rodpai et al., 2022), ref.no. 22.

IgG is the immunological marker used to measure CCA risk. I understand that the author attempted to mimic a scenario in which each specimen's infection status is confirmed. Therefore, the validity and performance of the kit may be evaluated. However, the author should be aware that this is not typical of an infection in which all infected individuals are at the same stage.

Reply: We agree with the reviewer comment that “this is not typical of an infection in which all infected individuals are at the same stage” however, the aims of this study are to support clinical diagnosis at the bedside but also for field-based surveys to predict risk of CCA of the positive IgG anti Opisthorchis in endemic and remote regions which may lack sophisticated instrumentation. The information is of note with respect to prognostic and diagnostic biomarkers for screening at risk development of CCA in regions endemic for liver fluke infection, and in disease prevention.

The experiment is engaging and instructive, but the author should be concerned and provide additional clarification, as I indicated for this issue.

I propose that referring to the kit as a tool for detecting Ov infection may be too optimistic for the reasons mentioned above.

Reply: The aims of this study are kit is to support clinical diagnosis at the bedside but also for field-based surveys to predict risk of CCA of the positive IgG anti Opisthorchis in endemic and remote regions which may lack sophisticated instrumentation. People in areas endemic for opisthorchiasis can be rapidly screened for liver fluke infection using our rapid kit, including where infections may be asymptomatic or mild, and where there are complications that can include hepatobiliary morbidity involving hepatomegaly, cholangitis, cholecystitis, periductal fibrosis and/or gallstones along with a stool negative examination finding for Opisthorchis eggs (Phupiewkham et al., High prevalence of opisthorchiasis in rural populations from Khammouane Province, central Lao PDR: serological screening using total IgG- and IgG4-based ELISA. Trans R Soc Trop Med Hyg. 2021 Dec 2;115(12):1403-1409. doi: 10.1093/trstmh/trab066.). The information is of note with respect to prognostic and diagnostic biomarkers for screening at risk development of CCA in regions endemic for liver fluke infection, and in disease prevention.

2. Test kit for diagnostic testing

Did the author intend to use this kit as a screening or diagnostic test?

This should be addressed in the manuscript. I assumed that the test scenario was designed for an Ov endemic environment, that the serum was collected from an Ov and Cs endemic region, and that simulated blood was enriched with lab-grown Ov antigens. With such a high infection density, the test could be a good candidate for a diagnostic test, not to mention its practicability and feasibility, which made it highly convenient.

Reply: The aims of this study are kit is to support clinical diagnosis at the bedside but also for field-based surveys to predict risk of CCA of the positive IgG anti Opisthorchis in endemic and remote regions which may lack sophisticated instrumentation. People in areas endemic for opisthorchiasis can be rapidly screened for liver fluke infection using our rapid kit, including where infections may be asymptomatic or mild, and where there are complications that can include hepatobiliary morbidity involving hepatomegaly, cholangitis, cholecystitis, periductal fibrosis and/or gallstones along with a stool negative examination finding for Opisthorchis eggs (Phupiewkham et al., High prevalence of opisthorchiasis in rural populations from Khammouane Province, central Lao PDR: serological screening using total IgG- and IgG4-based ELISA. Trans R Soc Trop Med Hyg. 2021 Dec 2;115(12):1403-1409. doi: 10.1093/trstmh/trab066.). The information is of note with respect to prognostic and diagnostic biomarkers for screening at risk development of CCA in regions endemic for liver fluke infection, and in disease prevention. For better understanding, we mentioned in the revised manuscript, conclusion.

In many regions of Thailand, the risk of infection and eating undercooked fish remains high. Despite this, the intensity of the infection is relatively low, which could result in a somewhat low amount of Ov antigen. Therefore, using this kit as a screening test may be problematic.

I encourage the author to elaborate on this subject in the article.

Reply: Thank for your advice, we agree with the reviewer comments and added the limitations of method in discussion part, please see revised manuscript, last paragraph, and conclusion, last sentence.

3. Predictive value

Positive and negative predictive value indicators are presented in this work.

The positive predictive value (PPV) is determined similarly to the risk ratio. Therefore, PPV will be associated with infection prevalence. As if the test were extensively utilized, PPV is considered the determining criterion for assessing cost-effectiveness by public health professionals.

However, this infection was simulated; consequently, we could not determine the true prevalence of Ov and Cs infection. As such, the PPV may not reflect the actual value. In contrast to sensitivity and specificity, they would be technically equivalent at all levels of disease prevalence.

Again, this is not wrong; the author should address this issue as it is still the outcome of an in-vitro study.

Reply: We revised as the reviewer comment, we added the sentence “However, this was an in vitro study with simulated WBS in a laboratory setting, more evaluation should be conducted in field studies, please see conclusion, last sentence.

4. Page 5

The explanations of sens/spec and PPV/NPV are redundant. The author could relocate the explanation to the appendix to shorten the document's main body.

Reply: We revised as suggested, please see revised manuscript, page 5. “2.6. Data analysis”.

Finally, we would like to thank the reviewer very much. Your comments are supportive and helpful.

Reviewer 2 Report

1. Does the band intensity is related to EPG

2. Reading IgG based detection, author should discuss about the window period of this test.

3. The previous version of the rapid detection kit used plasma as a sample. The plasma has to prepare before loafing into the ICT which limits the test kit for point of care test (POCT). This study develops a new kit that allows using the whole blood instead of plasma to serve the POCT.
4. This study is an innovative test for Opisthorchiasis and Clonorchiasis. The topic is original and it has never been reported.
5. The test kit for Opisthorchiasis has been never developed. Using this kit for screening of Opisthorchiasis is valuable and helpful in terms of the prevention of cholangiocarcinoma development
6. This manuscript is well-written and easy to read and understand.
7. The conclusion is consistent with the evidence and arguments. However, the cross-reactivity of this kit with Clonorchiasis is not specific to Opisthorchiasis and it should be broad detection for liver flukes.

Author Response

Point by point response to Reviewer 2 comments

Comments and Suggestions for Authors

1. Does the band intensity is related to EPG

Reply: The band intensity is not related to EPG, please see table S1. This results may be due to not all infected individuals are at the same stage where infection may be asymptomatic or mild, and where there are complications that can include hepatobiliary morbidity involving hepatomegaly, cholangitis, cholecystitis, periductal fibrosis and/or gallstones along with a stool negative examination finding for Opisthorchis eggs. Latent infection with low worm burdens or biliary obstruction cause a stool negative examination finding for Opisthorchis eggs. (Phupiewkham et al. High prevalence of opisthorchiasis in rural populations from Khammouane Province, central Lao PDR: serological screening using total IgG- and IgG4-based ELISA. Trans R Soc Trop Med Hyg. 2021 Dec 2;115(12):1403-1409. doi: 10.1093/trstmh/trab066.).

2. Reading IgG based detection, author should discuss about the window period of this test.

Reply: We mentioned in discussion, last paragraph.

3. The previous version of the rapid detection kit used plasma as a sample. The plasma has to prepare before loafing into the ICT which limits the test kit for point of care test (POCT). This study develops a new kit that allows using the whole blood instead of plasma to serve the POCT.

Reply: We appreciate your supportive remarks. We are delighted that you think our manuscript has significance.

4. This study is an innovative test for Opisthorchiasis and Clonorchiasis. The topic is original and it has never been reported.

Reply: Thank you very much.

5. The test kit for Opisthorchiasis has been never developed. Using this kit for screening of Opisthorchiasis is valuable and helpful in terms of the prevention of cholangiocarcinoma development.

Reply: Thank you very much.

6. This manuscript is well-written and easy to read and understand.

Reply: Thank you very much.

7. The conclusion is consistent with the evidence and arguments. However, the cross-reactivity of this kit with Clonorchiasis is not specific to Opisthorchiasis and it should be broad detection for liver flukes.

Reply: Thank you very much.

Reviewer 3 Report

The Authors propose an interesting innovation of an immuno-cromatographic test for the diagnosis of Opistorchis viverrini (and secondly Clonorchis sinensis) human infection, optimizing the method for the whole-blood samples instead the serum. The major interest of this adaptation would be the use of the kit as POC test in remote regions in absence of specialized laboratories or equipment. The research design sounds quite well even if many points of the text appear confused and hard-to-understand especially for a not correct use of the language (I suggest an extensive English editing).

Following some specific suggestions and comments:

-        Key words (and everywhere in the text): according to the international standardised nomenclature (see SNOAPAD  at https://www.waavp.org/documents/snopad-guidelines/#.YzwMtHZBxPY) it should be used the suffix “osis” when the name of diseaes is formed from the taxonomic name of parasite: so, change in “opistorchiosis” and “clonorchiosis”.

-        Line 7 page 1: clonorchiasis sera… the innovation of the work is the use of WBS and not sera as in the past version of the kit; change in accordance with that expressed in the article.

-        Line 8 page 2 (and everywhere in the text): the different use of acronyms WBS and WBSs generate a great confusion through all the article. Also here for example: talking in generically about the type of sample (serum or blood), the correct acronym would be WBS; use WBSs only for the simulated samples: you have to name these or “simulated WBS” or “simply WBSs”. Check all the test and change in accordance with this rule.

-        Last 4 lines in introduction, page 2: if you speak about the new test termed OvB-test delete these lines as they anticipate the results and the conclusion of your study; keep them if you speak generally about the ICT kit of Phupiewkam (that uses only serum), changing accordingly the name of the kit in the sentence.

-        Line 4 in Materials and Methods: “as described by…” (put the names of the Authors of the reference)

-        Line 11 in Materials and Methods: “was determined according to the method proposed by…” (explain by who or where the method for protein concentration have been used, not only the number of reference)

-        Line 4 in Clinical samples: sign in letters, and not numbers, the four groups…a), b), etc…in this way they are not confused with the bibliographic references.

-        Line 16 page 3: same problem for acronym WBS as over…It appears that you used WBS (not simulated) of other persons with confirmed opistorchiosis (Table S1) so these are WBS and not WBSs.

-        Page 3: Table 1 includes results, I suggest to move forward in the “Results” chapter.

-        Line 1 page 4 in chapter 2.4: (Figure A) and line 6 in chapter 2.5: (Figure B)

-        Figure 1: move (B) at the end of “reference board”. There are two positive test in the Figure: eliminate one or, if not, for example state at what intensity level of positivity are the two bands in T line.

-        Page 6, Figure 2: as in Figure 1, keep only one of the positive test about Opistorchiosis group, Healthy control group and Clonorchiosis group; if you want keep all, give at least (if  there are some interesting difference) the level of intensity in the band according to the reference card for giving some sense to these repetitive images. About the test images of “Other parasitic infection group” as they seem all negative (except Fascioliosis…why?), eliminate all as they don’t give a significant contribute to that yet well expressed in the text.

-        Line 1 in Discussion, page 6: develop the acronym POC in Point-of-Care (POC).

-        Line 6 in Discussion in page 6: …and do make use WBS…perhaps “don’t make use of WBS”? The aim of paper was that, proposing an innovation.

-        Line 3 page 7: …clonorchiasis sera? From the materials and method it seems you have make simulated WBS also for these: if it’s so, eliminate sera and describe the samples used in the kit according to opistorchiosis.

-        Line 15-17 page 7: it would be interesting give here some hypothesis about why whole blood samples of field (WBS) give band level between 0.5-2 (Tabel S1) unlike 0.5-5 simulated whole blood samples (WBSs) reported in Table 1. Explain the reasons for the Authors.

-        Line 28 page 7: check the references 37-44 as they don’t relate to “the clinically different setting in opistorchiosis and clonorchiosis”.

Author Response

Point by point response to reviewer 3

Comments and Suggestions for Authors

The Authors propose an interesting innovation of an immunocromatographic test for the diagnosis of Opisthorchis viverrini (and secondly Clonorchis sinensis) human infection, optimizing the method for the whole-blood samples instead the serum. The major interest of this adaptation would be the use of the kit as POC test in remote regions in absence of specialized laboratories or equipment. The research design sounds quite well even if many points of the text appear confused and hard-to-understand especially for a not correct use of the language (I suggest an extensive English editing).

Reply: Thank you very much for your kind suggestions. Your comments were clarified in the revised text. A language editing service (Language Editing Services, MDPI) corrected the grammar and improved the English in the revised manuscript.

Following some specific suggestions and comments:

-Key words (and everywhere in the text): according to the international standardised nomenclature (see SNOAPAD  at https://www.waavp.org/documents/snopad-guidelines/#.YzwMtHZBxPY) it should be used the suffix “osis” when the name of diseaes is formed from the taxonomic name of parasite: so, change in “opistorchiosis” and “clonorchiosis”.

Reply: Thank you for your suggestion. Due to this experiment involved human infectious diseases, “opisthorchiasis” and “clonorchiasis” were widely used (please see https://pubmed.ncbi.nlm.nih.gov/?term=Opisthorchiasis&format=abstract&sort=date) and (please see https://pubmed.ncbi.nlm.nih.gov/?term=clonorchiasis&sort=date). We would like to use these words in the present publication. We intend these words will be familiar with clinicians and other readers who work in clinical practices. This is our reason.

-Line 7 page 1: clonorchiasis sera… the innovation of the work is the use of WBS and not sera as in the past version of the kit; change in accordance with that expressed in the article.

Reply: We modified thoroughly the texts as you suggested, line 6, page 7, discussion.

-Line 8 page 2 (and everywhere in the text): the different use of acronyms WBS and WBSs generate a great confusion through all the article. Also here for example: talking in generically about the type of sample (serum or blood), the correct acronym would be WBS; use WBSs only for the simulated samples: you have to name these or “simulated WBS” or “simply WBSs”. Check all the test and change in accordance with this rule.

Reply: We modified thoroughly the texts as you suggested.

-Last 4 lines in introduction, page 2: if you speak about the new test termed OvB-test delete these lines as they anticipate the results and the conclusion of your study; keep them if you speak generally about the ICT kit of Phupiewkam (that uses only serum), changing accordingly the name of the kit in the sentence.

Reply: We modified thoroughly the texts as you suggested.

-Line 4 in Materials and Methods: “as described by…” (put the names of the Authors of the reference)

Reply: We modified as you suggested.

-Line 11 in Materials and Methods: “was determined according to the method proposed by…” (explain by who or where the method for protein concentration have been used, not only the number of reference)

Reply: We modified as you suggested.

-Line 4 in Clinical samples: sign in letters, and not numbers, the four groups…a), b), etc…in this way they are not confused with the bibliographic references.

Reply: We modified as you suggested.

-Line 16 page 3: same problem for acronym WBS as over…It appears that you used WBS (not simulated) of other persons with confirmed opistorchiosis (Table S1) so these are WBS and not WBSs.

Reply: We modified as you suggested.

-Page 3: Table 1 includes results, I suggest to move forward in the “Results” chapter.

Reply: We modified as you suggested.

-Line 1 page 4 in chapter 2.4: (Figure A) and line 6 in chapter 2.5: (Figure B)

Reply: We modified as you suggested.

-Figure 1: move (B) at the end of “reference board”. There are two positive test in the Figure: eliminate one or, if not, for example state at what intensity level of positivity are the two bands in T line.

Reply: We modified as you suggested, please see revised Figure 1.

-Page 6, Figure 2: as in Figure 1, keep only one of the positive test about Opistorchiosis group, Healthy control group and Clonorchiosis group; if you want keep all, give at least (if  there are some interesting difference) the level of intensity in the band according to the reference card for giving some sense to these repetitive images. About the test images of “Other parasitic infection group” as they seem all negative (except Fascioliosis…why?), eliminate all as they don’t give a significant contribute to that yet well expressed in the text.

Reply: We modified as you suggested, please see revised Figure 2. The positive band in fascioliasis serum may be due to cross-reaction. Please see Table 1.

-Line 1 in Discussion, page 6: develop the acronym POC in Point-of-Care (POC).

Reply: We added in introduction part, lines 9-10.

-Line 6 in Discussion in page 6: …and do make use WBS…perhaps “don’t make use of WBS”? The aim of paper was that, proposing an innovation.

Reply: We modified as you suggested.

-Line 3 page 7: …clonorchiasis sera? From the materials and method it seems you have make simulated WBS also for these: if it’s so, eliminate sera and describe the samples used in the kit according to opistorchiosis.

Reply: We modified to” human clonorchiasis” as you suggested. Please see discussion, page 7, line 9.

-Line 15-17 page 7: it would be interesting give here some hypothesis about why whole blood samples of field (WBS) give band level between 0.5-2 (Tabel S1) unlike 0.5-5 simulated whole blood samples (WBSs) reported in Table 1. Explain the reasons for the Authors.

Reply: We modified as you suggested, please see revised version discussion, last paragraph. We added the sentences “Third, clinicians working in this endemic area should be reminded that there were differences in the reactions of detection, as the WBS from the field collections gave a band level between 0.5 and 2 (Table S1), and the band level was between 0.5 and 5 in the simulated WBS reported in Table 1. These results may be due to sample collections from different period of infections or the intensity of the helminthic infections in the populations studied.”

-Line 28 page 7: check the references 37-44 as they don’t relate to “the clinically different setting in opistorchiosis and clonorchiosis”.

Reply: We modified as you suggested we change ref. no. 40 from “Yamasaki, H. Current status and perspectives of cysticercosis and taeniasis in Japan. Korean. J. Parasitol. 2013, 51, 19-29. https://doi.org/10.3347/kjp.2013.51.1.19” to “Garcia, H.H.; Nash, T.E.; Del Brutto, O.H. Clinical symptoms, diagnosis, and treatment of neurocysticercosis. Lancet Neurol. 2014, 13, 1202-1215. doi: 10.1016/S1474-4422(14)70094-8.”

Finally, We would like to thank you very much for your comments, your comments are very supportive and helpful.

Reviewer 4 Report

In this study Sadaow et al have developed a lateral flow-based test for the detection of Opisthorchis viverrini infections in humans via direct analysis of whole blood samples. The test is rapid and requires no sophisticated laboratory equipment, and as such, will be of particular use in rural areas in endemic regions.

The test appears to perform very well however there are some problems with cross-reactivity to other parasite species which could lead to misdiagnosis/treatment.  Have the authors considered preparing a test using purified Ov recombinant molecules that are known to be secreted/antigenic?  Since the whole fluke extract contains >30 proteins this approach may reduce the likelihood of cross-reactivity. The wider Khon Kaen Ov lab have produced a variety of quality recombinants over the years that could be tested.  

Author Response

Point by point response to Reviewer 4

Comments and Suggestions for Authors

In this study Sadaow et al have developed a lateral flow-based test for the detection of Opisthorchis viverrini infections in humans via direct analysis of whole blood samples. The test is rapid and requires no sophisticated laboratory equipment, and as such, will be of particular use in rural areas in endemic regions.

The test appears to perform very well however there are some problems with cross-reactivity to other parasite species which could lead to misdiagnosis/treatment.  Have the authors considered preparing a test using purified Ov recombinant molecules that are known to be secreted/antigenic?  Since the whole fluke extract contains >30 proteins this approach may reduce the likelihood of cross-reactivity. The wider Khon Kaen Ov lab have produced a variety of quality recombinants over the years that could be tested. 

Reply: Thank you for your encouraging comment and for pointing this out. The lateral flow-based test using purified Opisthorchis viverrini recombinant molecules for the detection of O. viverrini infections in humans is in the production and evaluation way. However, this is the next experiment.

Round 2

Reviewer 1 Report

I do not have any additional comments. The revised version is acceptable.